# Segmenting Ischemic Penumbra and Infarct Core Simultaneously on Non-Contrast CT of Patients with Acute Ischemic Stroke Using Novel Convolutional Neural Network

**DOI:** 10.3390/biomedicines12030580

**Published:** 2024-03-05

**Authors:** Hulin Kuang, Xianzhen Tan, Jie Wang, Zhe Qu, Yuxin Cai, Qiong Chen, Beom Joon Kim, Wu Qiu

**Affiliations:** 1Hunan Provincial Key Lab on Bioinformatics, School of Computer Science and Engineering, Central South University, Changsha 410083, China; hulinkuang@csu.edu.cn (H.K.); 224712155@csu.edu.cn (X.T.); jwang@csu.edu.cn (J.W.); zhe_qu@csu.edu.cn (Z.Q.); 2School of Life Science and Technology, Huazhong University of Science and Technology, Wuhan 430074, China; m202272350@hust.edu.cn; 3Ultrasound Diagnosis Department, Wuhan No. 1 Hospital, Wuhan 430022, China; qiongchen.wh@gmail.com; 4Department of Neurology, Seoul National University Bundang Hospital, Seongnam-si 13620, Republic of Korea; kim.bj.stroke@gmail.com; 5Gyeonggi Regional Cerebrovascular Center, Seoul National University Bundang Hospital, Seongnam-si 13620, Republic of Korea

**Keywords:** acute ischemic stroke, ischemic penumbra and ischemic core segmentation, non-contrast CT, multi-scale convolution, symmetry enhancement, hierarchical deep supervision

## Abstract

Differentiating between a salvageable Ischemic Penumbra (IP) and an irreversibly damaged Infarct Core (IC) is important for therapy decision making for acute ischemic stroke (AIS) patients. Existing methods rely on Computed Tomography Perfusion (CTP) or Diffusion-Weighted Imaging–Fluid Attenuated Inversion Recovery (DWI-FLAIR). We designed a novel Convolutional Neural Network named $I^{2}$PC-Net, which relies solely on Non-Contrast Computed Tomography (NCCT) for the automatic and simultaneous segmentation of the IP and IC. In the encoder, Multi-Scale Convolution (MSC) blocks were proposed to capture effective features of ischemic lesions, and in the deep levels of the encoder, Symmetry Enhancement (SE) blocks were also designed to enhance anatomical symmetries. In the attention-based decoder, hierarchical deep supervision was introduced to address the challenge of differentiating between the IP and IC. We collected 197 NCCT scans from AIS patients to evaluate the proposed method. On the test set, $I^{2}$PC-Net achieved Dice Similarity Scores of 42.76 ± 21.84%, 33.54 ± 24.13% and 65.67 ± 12.30% and lesion volume correlation coefficients of 0.95 (*p* < 0.001), 0.61 (*p* < 0.001) and 0.93 (*p* < 0.001) for the IP, IC and IP + IC, respectively. The results indicated that NCCT could potentially be used as a surrogate technique of CTP for the quantitative evaluation of the IP and IC.

## 1. Introduction

Acute ischemic stroke (AIS) is caused by the occlusion or blockage of small or large blood vessels due to a thrombus or embolism event, resulting in reduced blood flow to a portion of the brain tissue. It accounts for 87% of all strokes and has high morbidity and mortality [1,2]. Once an AIS occurs, a portion of the brain tissue may have already suffered irreversible damage (the Infarct Core, IC), and the surrounding brain tissue is also at risk due to reduced blood flow (the Ischemic Penumbra, IP) and may be salvageable [3,4]. Therefore, the goal of AIS treatment is to reperfuse the blood-deprived area before the salvageable IP transforms into the IC. The treatment methods for AIS patients mainly include intravenous thrombolysis and endovascular therapy [5]. Neuroradiologists usually select the appropriate treatment method for patients based on clinical guidelines, e.g., mechanical thrombectomy being more suitable when the IC volume is less than 70 mL, the IP volume is greater than 15 mL and the IP to IC ratio exceeds 1.8 [5,6,7,8]. Due to the extremely short 4.5 h treatment window, the rapid and accurate assessment of the volume and location of the IP and IC is important for reperfusion therapy decision making for AIS patients.

In clinical practice, neuroradiologists typically evaluate the IP and IC through manual delineation on multi-modal images, such as by using diffusion imaging to identify the IC and diffusion–perfusion mismatch to identify the IP. However, these manual segmentations are subject to interobserver and intraobserver variability and fatigue-related errors, and they are time consuming. Moreover, invasive imaging modalities are sometimes unavailable. Therefore, a rapid, objective, accurate and widely applicable method for automated IP and IC segmentation is desired in the computer-aided diagnosis of AIS.

Machine learning and deep learning methods have been extensively used in recent years for fully automatic medical image segmentation. Numerous general 3D medical image segmentation methods are available for the segmentation of the IP and IC, such as [9,10,11,12,13,14], etc. Additionally, some researchers have developed specialized machine learning and deep learning methods for infarct lesion segmentation. Gupta et al. [15] designed a U-shaped encoder–decoder network named MSNet. They utilized a combination of eight modalities of diffusion and perfusion maps to segment the IP and IC, where the diffusion–perfusion mismatch facilitates the differentiation between the IP and IC. Bhurwani et al. [16] utilized U-Net [17] to segment the IC and IP + IC from CTP scans, but they did not differentiate between the IP and IC. Lee et al. [18] and Vupputuri et al. [19] both adopted Diffusion-Weighted Imaging–Perfusion-Weighted Imaging (DWI-PWI) to quantify and differentiate the IP and IC. Werdiger et al. [20] explored XGBoost, followed by 3D neighborhood analysis, for the concurrent segmentation of the IP and IC on CTP scans. Tomasetti et al. [21] implemented a 4D Convolutional Neural Network (CNN) approach to leverage the spatiotemporal data contained within CTP scans, thereby delineating the IP and IC. Sathish et al. [22] deployed an adversarially trained CNN to segment the IP and IC simultaneously from multi-sequence Magnetic Resonance Imaging (MRI) scans. In summary, the specialized methods mentioned here either do not strictly differentiate between the IP and IC or they rely on multiple advanced imaging modalities such as CTP, PWI and DWI to differentiate the IP and IC. However, these advanced imaging techniques are time consuming and sometimes even unavailable, and fast and cheap Non-Contrast CT (NCCT) has seldom been considered in previous studies.

In this study, we propose a neural network named $I^{2}$PC-Net, which relies solely on widely available, cheap and fast baseline NCCT scans to simultaneously segment the IP and IC. $I^{2}$PC-Net has a seven-level U-shaped encoder–decoder architecture, relying on pure convolution. In the encoder, to model the varying shapes, sizes and locations of the infarct lesions, we designed the Multi-Scale Convolution (MSC) block. To model the anatomical symmetry and capture the difference between the left and right sides of the brain, we propose the Symmetry Enhancement (SE) block. In the attention-based decoder, we utilized hierarchical deep supervision mechanisms for the entire ischemic region (IP + IC) in the three deep levels and for differentiating the IP and IC at the three low levels. Through the effective strategies proposed above, we hypothesized that the $I^{2}$PC-Net can segment the IP and IC from NCCT well. Our contributions are summarized as follows: (1) We propose the MSC block to model the high variability of AIS lesions. (2) We introduce the SE block to capture the differences between the bilateral hemispheres of the brain. (3) An attention-based decoder was employed to better integrate high-level and low-level features. (4) Hierarchical deep supervision was designed to more effectively differentiate between the IP and IC on NCCT.

## 2. Materials and Methods

### 2.1. Data Acquisition

We collected multi-modal data including DWI, Fluid-Attenuated Inversion Recovery (FLAIR) and NCCT from 197 AIS patients in a prospective stroke registry at a single academic center. The institutional review board of the Seoul National University Bundang Hospital approved the data analysis, image evaluation and modeling process (B-2102/667-106). The included patients or their next of kin provided written consent for the prospective clinical stroke registry to record and collect their data (B-1401/236-007, B-1706/403-303).

All the modalities were coregistered to NCCT. There were two inclusion criteria for the patient samples: (1) each modality’s data encompass the entire brain without significant artifacts, and (2) expert annotations of the ischemic tissue region are available. In the dataset, the number of slices in the sagittal view is 512 and the ranges of the number of slices in the coronal and axial views are 512–638 and 28–37, respectively. The range of spacing is 0.326–0.429 mm for both the sagittal and coronal views and 4.999–5.015 mm for the axial view. Groundtruth labels for the IC and IP on NCCT were defined by high signal regions on DWI and DWI-FLAIR mismatch areas, respectively. These labels were first annotated by a neuroradiologist (Qiong Chen) with over 5 years of experience using the software ITK-SNAP version 4.2.0. [23] and were then double checked by another neuroradiologist (Beom Joon Kim) with over 10 years of experience to achieve accurate annotations. Finally, we utilized these 197 annotated NCCT scans and divided them in a ratio of 7:2:1 for training, validation and testing, respectively.

### 2.2. Image Preprocessing

To eliminate the influence of the skull region, we first removed the skull following the method proposed by Najm et al. [24]. Figure 1 sequentially displays the NCCT after the skull removal, DWI with highlighted infarct signals, FLAIR showing a mismatch with DWI and the category labels for the IP (red) and IC (green).

Considering the robust and powerful performance of nnUNet [9] for medical image segmentation, we followed its preprocessing approach, which depends on the statistical information of a specific dataset (called the dataset fingerprint). Initially, the images were cropped based on the 3D bounding box of the brain tissue to avoid unnecessary computations. Subsequently, all the images were resampled to the dataset’s median voxel spacing: 0.3789 mm × 0.3789 mm × 5.0 mm. This enhances the performance of CNN networks with inductive bias, enabling them to better learn the typical sizes of brain anatomical structures. Finally, Z-Score normalization was performed based on the mean and variance of the segmentation target (take pixel values within the range of 0.5% to 99.5%). This is equivalent to considering the window width and window level of the target lesion or anatomical structure, which helps the network learn more effective features and accelerates convergence.

### 2.3. The Proposed $I^{2}$PC-Net

As illustrated in Figure 2a, $I^{2}$PC-Net also adopted a U-shaped structure with a 7-level encoder and a 6-level decoder. The feature channel (i.e., the number of convolution filters) of each encoder level and decoder level are also given in Figure 2a. The three low encoder levels were composed of MSC blocks, and the four deep encoder levels added an SE block after the MSC block. Convolution-based downsampling was interleaved between two adjacent encoder levels. In the six-level decoder, we adopted the attention-based decoder of Oktay et al. [25] to better fuse high-level semantic information with low-level fine-grained image details. Transposed convolution-based upsampling was interleaved between two adjacent decoder levels. Note that for the spatial dimension D, downsampling or upsampling by a convolution of stride 2 was performed twice, i.e., only in the two levels above the bottleneck. Whereas for the H and W dimensions, upsampling or downsampling by a convolution of stride 2 occurred at every level. The input stem and segmentation head were, respectively, responsible for initial feature embedding and output generation. Considering the difficulty in differentiating the IP and IC on NCCT, a hierarchical deep supervision decoding mechanism was used for the decoder levels.

#### 2.3.1. Multi-Scale Convolution Block

Existing general 3D medical image segmentation methods such as those targeting abdominal multi-organ segmentation and the similar shape size and location of the organs determine the feasibility of single-scale modeling. However, high variability in the location, size and shape of the infarct lesions needs multi-scale modeling. Inspired by Guo et al. [26], we propose the MSC block, as shown in Figure 2b. For the sake of simplicity in the diagram, we omitted the activation functions and normalization operations. Taking the MSC block in the first level of the encoder for example, the feature map was firstly passed to a vanilla convolution block (convolution with a kernel size of 3 × 3 × 3 + Instance Normalization + LeakyReLU). The output of this convolution operation was also added to the final output as a residual connection. Then, we designed a parallel depth-wise convolution branch with kernel sizes of 5, 7 and 11 to obtain the multi-scale features (note that here, all depth-wise convolutions were followed neither by normalization nor by activation functions, and to further enlarge the receptive field, a 3 × 3 × 3 depth-wise convolution was positioned before the other three scales). The four outputs of multiple depth-wise convolutional branches were added element-wise and then passed through a fusion convolution block (convolution with a kernel size of 1 × 1 × 1 + Instance Normalization + LeakyReLU). Additionally, to reduce the complexity of the model, depth-wise convolutions with a kernel size of 1 × k × k followed by a kernel size of k × 1 × 1 were employed in place of a kernel size of k × k × k, where k∈{5,7,11}. Given an input feature map $X\in R^{B\times C\times D\times H\times W}$, the MSC block’s output feature map $Y\in R^{B\times C\times D\times H\times W}$ can be formalized as follows:(1)$$X^{\prime }=\sigma \left(\mathrm{Norm}\left({\mathrm{Conv}}_{3\times 3\times 3}\left(X\right)\right)\right)$$
(2)$$Y={\mathrm{Conv}}_{1\times 1\times 1}\left(\sum_{i=0}^{3}{\mathrm{Scale}}_{i}\left(\mathrm{DWConv}\left(X^{\prime }\right)\right)\right)+X^{\prime }$$
where ${\mathrm{Conv}}_{k\times k\times k}$ denotes the convolution with a kernel size of k×k×k, Norm represents Instance Normalization, σ is the LeakyReLU activation function and ${\mathrm{DWConv}}_{k\times k\times k}$ indicates a depthwise convolution with a kernel of k×k×k. ${\mathrm{Scale}}_{i}\left(\mathrm{DWConv}\left(\cdot \right)\right)$ represents the *i*-th depthwise convolutional branch, where i = 0 indicates the identity connection.

#### 2.3.2. Symmetry Enhancement Block

The left and right hemispheres of the brain exhibit axial symmetry along the mid-sagittal line. Typically, the opposite side of a cerebral infarction is normal brain tissue. Previous studies had utilized this prior clinical knowledge to enhance the model’s ability to locate suspicious ischemic lesions. However, due to variations in patient positioning during imaging, the mid-sagittal line in the image may not be vertical. Previous strategies include the use of alignment neural networks and direct registration [27,28,29,30,31,32,33]. We believe that the influence of slight tilts in brain scans can be mitigated at higher semantic levels, where each pixel represents a larger area of the original image. Therefore, we directly appended an SE block after the MSC block in the 4 high levels of the encoder. The structure of the SE block is shown in Figure 2c. The feature map from the MSC block was first horizontally flipped and then it was element-wise subtracted from the flipped feature maps. The obtained feature map after subtraction was concatenated with the input feature map along the channel dimension. Subsequently, it passed through a convolution block (convolution with a kernel size of 1×1×1 + Instance Normalization + LeakyReLU) to obtain the fused feature. Finally, the fused feature map was element-wise added to the input feature map to produce the final output. Given the input feature map $H\in R^{B\times C\times D\times H\times W}$ from the MSC block, the SE block’s output $H_{\mathrm{SE}}\in R^{B\times C\times D\times H\times W}$ can be formulated as
(3)$$H_{\mathrm{SE}}={\mathrm{Conv}}_{1\times 1\times 1}\left(\mathrm{Concat}\left(H,H-H_{\mathrm{flipped}}\right)\right)+H$$
where $H_{\mathrm{flipped}}$ represents the feature map after horizontal flipping and Concat denotes concatenation along the channel dimension.

#### 2.3.3. Attention-Based Decoder

The rational fusion of coarse-grained and fine-grained features is important for the final segmentation output. AttnUNet [25] introduced gated attention units in skip connections. It used coarse-grained feature maps as queries to weight fine-grained feature maps from the same level encoder, thereby learning which spatial regions to focus on. Because this structure was designed to address anatomical structures with highly variable shapes, we believed that this design was equally applicable to stroke segmentation. In Figure 2, it was denoted as “Attn Decoder Block”. Specifically, for each level of the “Attn Decoder Block”, the features from its subsequent level and the features from the corresponding level of the encoder were passed through a 1×1×1 convolution layer (the number of channels was halved) and then added element-wise. This was followed by a ReLU activation function and then another 1×1×1 convolution layer (where the number of channels was reduced to 1). The output then went through a Sigmoid activation function to obtain the weight (spatial attention score) at each pixel position. Finally, these weights were used to element-wise multiply with the features from the skip connections, thereby suppressing irrelevant feature responses in the fine-grained feature maps from the encoder. Lastly, the features from the subsequent level and the gated attention-modified features from the encoder at the same level were concatenated along the channel dimension and fused through a convolution layer. For detailed information, please refer to their publication [25]. We believe that, building upon the precise and more powerful encoding blocks like MSC and SA, those multi-scale, symmetry-enhanced features could better suppress irrelevant feature responses transmitted from skip connections, making the final features more effective.

#### 2.3.4. Hierarchical Deep Supervision

Owing to the exceedingly subtle differences between the IP and IC on NCCT, differentiating them directly in the deep layers of the network poses a significant challenge. In clinical practice, neuroradiologists initially approximate the location of the ischemic area and subsequently fine-tune the entire ischemic regions into the IP and IC. Drawing inspiration from this, we incorporated a hierarchical deep supervision strategy. Firstly, we continuously downsampled the ground truth label to match the spatial resolution of each decoder level. For each decoder level, we used a 1×1×1 convolutional layer to change the number of channels to the number of classification categories to achieve segmentation. That was, for the higher three levels, the number of categories was 2 (the background and IP + IC), and in the lower three levels, the number of categories was three (the background, IP and IC). We then calculated the loss by using the outputs of different levels and the corresponding downsampled ground truth. We used a linear combination of the Dice Similarity Coefficient (DSC) loss and Cross-Entropy (CE) loss as the objective function for each decoder level: $L=\alpha L_{DSC}+\beta L_{CE}$, where α and β are set to 1 in our practice. The total objective function of hierarchical deep supervision can be formalized as
(4)$$L_{\mathrm{Total}}=\sum_{i=0}^{2}Res_{i}\times L_{IP+IC}+\sum_{i=3}^{5}Res_{i}\times \left(L_{IP}+L_{IC}\right)$$
where $L_{IP+IC}$ represents the loss for the total ischemic area, treating the IP and IC as a single category, while $L_{IP}$ and $L_{IC}$, respectively, denote the losses for the IP and IC regions and $Res_{i}$ represents the weights of different decoder level’s supervision loss. When *i* ranges from 0 to 5 (six decoder levels from bottom to top), $Res_{i}$ takes the respective values of 0.02, 0.08, 0.2, 0.1, 0.2 and 0.4.

### 2.4. Implementation Details

We randomly sampled 3D patches of size 20 × 320 × 256 from the resampled and normalized data. For each patch, data augmentation includes spatial transformation (random rotation, random scaling and random elastic deformation), mirror transformation, adding white Gaussian noise, Gaussian blurring, low-resolution simulation, Gamma transformation and contrast and brightness adjustments. An initial learning rate of $1\times 10^{-2}$ with a polynomial decay schedule and a batch size of 2 were used. The Stochastic Gradient Descent (SGD) optimizer with a Nesterov Momentum of 0.99 and weight decay of $2\times 10^{-5}$ was used. The gradient clipping was set during training. We trained for 300 epochs, whereby each epoch consisted of 250 iterations. The code is available at https://github.com/GitHub-TXZ/I2PC-Net/, which is accessible to anyone for free, allowing for the validation and utilization of our method.

We adopted the sliding window strategy and Test Time Augmentation (TTA) strategy [9]. The window size is the same with the training patch size and its stride is 0.5× the patch size. The overlapping regions are weighted by a prepared Gaussian importance map. TTA is implemented via flipping along all axes. We did not perform any post-processing operations.

We compared several existing generic 2D or 3D segmentation methods, including pure CNN models such as nnUNet [9] and AttnUNet [25], pure Transformer models like nnFormer [12] and D-Former [11] as well as hybrid CNN and Transformer models like CoTr [13] and Swin-UNETR [14]. All the comparison methods were subjected to the same data processing and experimental settings to ensure fairness in the comparison. All the experiments were conducted on a ubuntu server (version 18.04) equipped with 5 NVIDIA A6000 48GB GPUs. The primary software dependencies include Pytorch version 2.0, nnU-Net version 2.2 (https://github.com/MIC-DKFZ/nnUNet (accessed on 20 December 2023)), MONAI version 1.3 (https://github.com/Project-MONAI/MONAI (accessed on 20 December 2023)) and Python version 3.10.11 (https://www.python.org/ (accessed on 20 January 2023)).

### 2.5. Statistical Analysis

In evaluating the segmentation performance for the IP, IC and IC + IP, we computed the metrics DSC, 95th percentile Hausdorff Distance (HD95) and Average Symmetric Surface Distance (ASSD) along with their respective means and standard deviations [31,34]. To assess the volume concordance between the manual segmentation made by neuroradiologists and the $I^{2}$PC-Net, we calculated Pearson’s correlation coefficients with a 95% confidence interval (CI) and generated regression and Bland–Altman plots. Given a 70 mL cut-off as the volume threshold for binary classification, the volume classification performance was evaluated by using accuracy, Area Under the Curve (AUC), Kappa and their respective 95% CIs. The statistical analyses were conducted by using MedCalc software (version 20.218, MedCalc Software Ltd., Mariakerke, Belgium) and the Python programming language (version 3.10.11, https://www.python.org/, (accessed on 20 January 2023)). *t*-test and proportion tests were used and a two-sided alpha level of less than 0.05 was considered to denote statistical significance.

## 3. Results

### 3.1. Study Participants

In the dataset comprising 197 collected cases, the median age of the research participants was 72 [IQR, 63–80], with 72 male subjects (57.36%). The median Onset-to-CT time was 73 [IQR, 41–180] min, and the median baseline NIHSS was 11 [IQR: 6–17]. The details of patient characteristics for all 197 AIS patients collected were listed in Table 1.

### 3.2. Results for IP Segmentation and IC Segmentation

We conducted comparisons with several existing 2D and 3D methods. Table 2 demonstrates the segmentation results for the IP segmentation and IC segmentation. From Table 2, our proposed $I^{2}$PC-Net achieved DSCs of 42.76% ± 21.84% and 33.54% ± 24.13%, HD95s of 13.81 ± 10.39 mm and 21.02 ± 14.81 mm and ASSDs of 3.59 ± 2.25 mm and 5.85 ± 4.28 mm for the IP and IC segmentation, respectively, outperforming all the compared 2D and 3D methods. These results show that our $I^{2}$PC-Net, benefiting from the effectiveness of the MSC and SA blocks, and the hierarchical deep supervision, achieved the optimal performance across various metrics. Overall, we could find that (1) the 3D methods were not necessarily superior to the 2D methods, which may be attributed to the large slice thickness, resulting in less strong connections between adjacent slices; (2) pure CNN approaches, such as AttnUNet [25] and nnUNet [9], continued to exhibit a robust performance in this task; (3) methods based solely on Transformers showed a weaker performance, possibly due to the challenges that Transformers face in smaller datasets rather than inherent limitations in the model itself; and (4) hybrid CNN–Transformer methods performed intermediately between pure Transformers and pure CNN methods. In other words, CNNs were more suitable for this task.

Figure 3 illustrates the visual segmentation results for our method and three representative methods: nnUNet [9], AttnUNet [25] and CoTr [13] for the IP and IC segmentation. In the figure, we could see that our $I^{2}$PC-Net could accurately locate the affected regions in the GTs of the IP and IC well, and also match the GT labels (DSC = 47.44% and 74.57% for the IP and IC, respectively) better than the three compared methods, showing its potential to provide affected-region information in clinical applications.

### 3.3. Results for the Entire Infarct (IP + IC) Segmentation

In clinical practice, the evaluation of the entire ischemic infarct (the IP + IC) is also of paramount importance for diagnosis and prognosis [31,43]. Therefore, we also evaluated the segmentation performance of the entire ischemic infarct. Without any additional training, all the methods’ segmentation results and the Groundtruth segmentation results treated the IP and IC labels as one category, without distinguishing between the IP and IC. Subsequently, the segmentation metrics were calculated to obtain Table 3. For the entire ischemic infarcts, our method achieved a DSC of 65.67% ± 12.30%, an HD95 of 12.54 ± 7.96 mm and an ASSD of 2.88 ± 1.27 mm, surpassing all comparative methods. In terms of the DSC, our method outperformed the best 2D method 2D nnUNet [9] (65.67% ± 12.30% vs. 49.44% ± 23.63%), the best pure 3D CNN method 3D nnUNet [9] (65.67% ± 12.30% vs. 62.22% ± 11.74%), the best pure 3D Transformer method D-former [11] (65.67% ± 12.30% vs. 45.18% ± 23.25%) and the best hybrid CNN–Transformer method CoTr [13] (65.67% ± 12.30% vs. 50.09% ± 22.14%).

As shown in the sixth subfigure (denoted by $I^{2}$PC-Net) in Figure 3, our method could accurately locate the entire ischemic regions and match the GT IP + IC well. Our method shows a similar DSC performance to nnUNet (89.17% vs. 89.77%) for the IP + IC segmentation. However, our method achieved a higher DSC for the IP segmentation and IP segmentation, showing its effectiveness at distinguishing the IP and IC.

**Figure 3 biomedicines-12-00580-f003:**
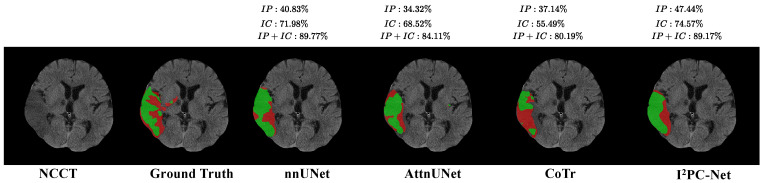
Visual segmentation results of the $I^{2}$PC-Net and three state-of-the-art methods: nnUNet, AttnUNet and CoTr. The red and green regions in each subfigure represent manual Groundtruth of IP and IC or IP and IC segmented by each compared algorithm, respectively. The numbers above the figure denote the DSC for IP, IC and IP + IC in this slice, respectively.

### 3.4. Volumetric Analysis of Segmented Infarcts

In clinical practice, the volume correlation as well as the infarct volume (e.g., 70 mL as the cut-off) are crucial for selecting AIS patients who will obtain good outcomes after different treatments [31,44,45]. Therefore, we also conducted a volume analysis on the ischemic infarcts obtained by our method to illustrate the clinical relevance of the proposed method.

Figure 4a–c illustrate the correlation analysis between the $I^{2}$PC-Net segmented volumes and manual segmented volumes for the IP, IC and IP + IC, respectively. The proposed $I^{2}$PC-Net achieved Pearson linear correlation coefficients (r) of 0.95 (95% CI: 0.9019–0.9720, *p* < 0.001), 0.61 (95% CI: 0.3637–0.7721, *p* < 0.001) and 0.93 (95% CI: 0.8749–0.9639, *p* < 0.001) for the IP, IC and IP + IC, respectively. These indicate a strong positive volume correlation for the IP and IP + IC, and the more challenging IC also exhibits a moderate volume correlation. Segmenting AIS infarcts in NCCT scans presents significant difficulties. First, compared to other imaging techniques like MRI, NCCT proves harder to analyze because of the lower signal-to-noise and contrast-to-noise ratios in cerebral tissues. Second, distinguishing infarct areas is complicated by normal physiological alterations, with the affected brain regions often exhibiting only slight differences in density and texture [32,33]. In the early stages of stroke, the IC does not appear significantly on NCCT, making it very difficult to distinguish between the IP and IC. Therefore, the correlation of the IC volume is relatively weak.

We also dichotomized the entire ischemic region (IP + IC) volume by using 70 mL as a cut-off and then evaluated the binary volume classification performance. Our developed $I^{2}$PC-Net demonstrated the capability to discriminate between patients with lesion volumes of ≤70 mL and >70 mL with a Kappa of 0.7536 [95% CI: 0.5579–0.9494], an AUC of 0.886 [95% CI: 0.746–0.965] and an accuracy of 87.50% [95% CI: 73.19%–95.81%], suggesting reasonable dichotomization volume information for therapy decision making.

## 4. Discussion

In this study, we explored a fully automatic segmentation approach named $I^{2}$PC-Net to simultaneously segment the IP and IC from NCCT scans. By employing MSC blocks, SA blocks and hierarchical deep supervision mechanisms, the proposed $I^{2}$PC-Net demonstrated a superior performance compared to some existing methods.

A comparative analysis with other methods revealed that the pure Transformer-based methods exhibited the poorest performance. The hybrid methods showed performance improvements over the pure Transformers, and they did so at the expense of convergence speed and computational cost. Pure convolutional approaches were more suited for this task in terms of convergence speed and final performance. Our method outperformed the powerful nnUNet, attributable to its enhanced capability of muti-scale modeling, suspected ischemic area locating and IP and IC differentiating. The experiment results confirm our hypothesis: employing modules like MSC blocks and SA blocks allow for the better handling of the substantial variability in infarct shape, size and location, while a hierarchical deep supervision decoding mechanism more effectively addresses the challenges in distinguishing between the IP and IC. This study demonstrates the feasibility of using only NCCT for simultaneous quantitative assessments of the IP and IC. $I^{2}$PC-Net can provide valuable insights for neuroradiologists in making therapeutic decisions, laying the groundwork for future researchers to develop more effective and broadly applicable methods.

From the quantitative segmentation metrics, visual segmentation results and volume analysis results, our approach demonstrated the effective localization of the ischemic region. Moreover, the classification performance using a cut-off volume of 70 mL was also favorable. This implies that in clinical applications, our method, relying solely on NCCT, can furnish valuable information for decision making in AIS treatments. The average time for automatic segmentation using the trained model was 3.49 s per NCCT scan, significantly enhancing the diagnostic efficiency of neuroradiologists for AIS patients.

Furthermore, to explore whether there were relevant clinical factors influencing the volume classification performance using 70 mL as cut-off, we conducted subgroup analyses based on factors such as gender, age, NIHSS score and Onset-to-CT time. As depicted in Table 4, the classification performance for patients aged over 70 was significantly superior to those under 70, and for patients with an Onset-to-CT-time exceeding 180 min, the classification performance was significantly better than for those below 180 min. No significant differences were found in the gender and NIHSS subgroups. A subgroup analysis indicates that age and Onset-to-CT time are two clinical factors closely associated with segmentation and classification performance. The reason why segmentation and lesion volume classification are more effective when the onset-to-CT time ≥ 180 min is that the longer the Onset-to-CT time, the more stable the lesion changes become, and lesions become more contrasted against the healthy tissues, making them easier to be segmented. However, given that the golden treatment window for AIS is 4.5 h, it is generally recommended in clinical practice to perform an NCCT scan and decide the appropriate treatment as soon as possible to improve the success rate of the intervention. Therefore, the above results do not imply that we should wait until after 180 min to collect NCCT data for treatment. Future research can incorporate age and Onset-to-CT time into the modeling process to further improve accuracy.

This study also has several limitations. First, the sample size in this paper is limited, and there is no external validation cohort. In the future, we aim to collect more data to train models that are more effective and broadly applicable. Second, from the qualitative results, we can find that even though the model accurately locates the entire ischemic region, the segmentation performance of the IP and IC individually might not be optimal. How to better distinguish the IP and IC while maintaining the accuracy of the entire ischemic IP + IC region remains a topic for future work.

## 5. Conclusions

This study proposed a pure CNN-based method, termed $I^{2}$PC-Net, which relies solely on NCCT to simultaneously and automatically segment the IP and IC. It mitigates the challenges of significant variations in the size, location and shape of infarct lesions through multi-scale modeling and Symmetry Enhancement blocks. We also employed a hierarchical deep supervision decoding mechanism to address the difficulty in distinguishing between the IP and IC in the deep layer. The results indicate that $I^{2}$PC-Net can automatically and quantitatively assess the IP and IC with good localization of the affected regions, strong volume correlation and high dichotomized volume classification performance, potentially providing valuable infarct information for diagnosis and patient selection in clinical applications.

## Figures and Tables

**Figure 1 biomedicines-12-00580-f001:**
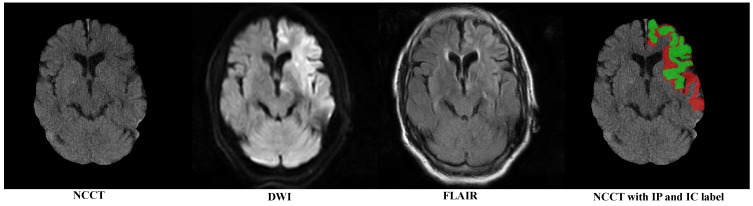
An example of multi-modal image of a patient; DWI and FLAIR are registered to NCCT. The red and green regions represent the manually annotated IP and IC, respectively.

**Figure 2 biomedicines-12-00580-f002:**
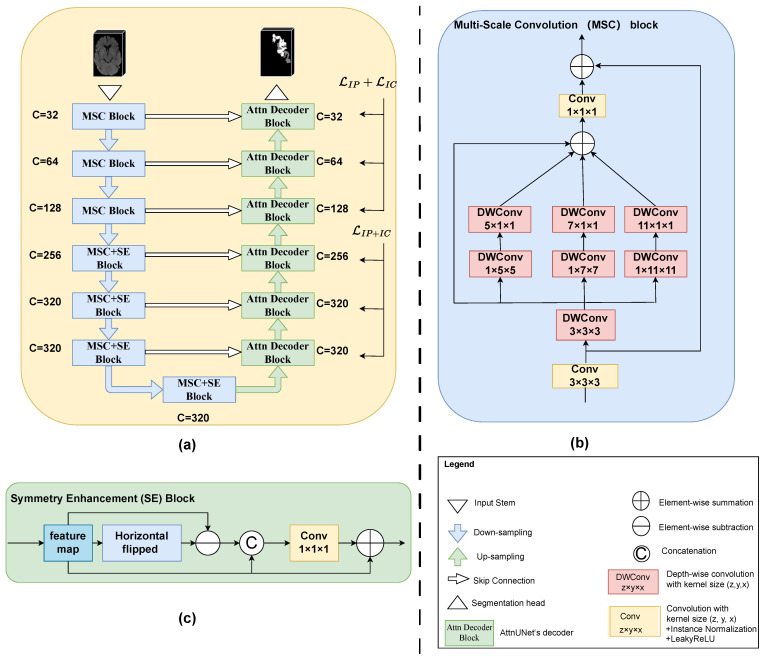
Architecture of the proposed $I^{2}$PC-Net. (**a**) Overview of the whole architecture. (**b**) Multi-Scale Convolution block. (**c**) Symmetry Enhancement block.

**Figure 4 biomedicines-12-00580-f004:**
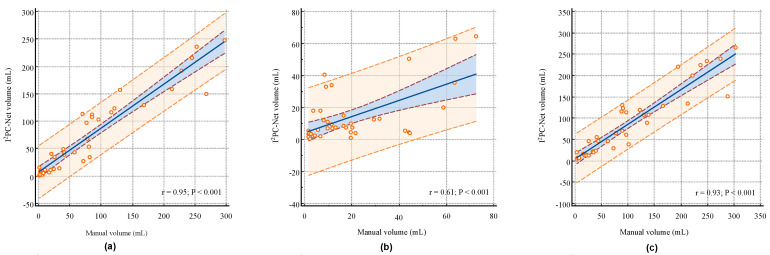
Volume correlation and consistency analysis of the segmented volume by $I^{2}$PC-Net compared with manual segmentation volumes. (**a**–**c**) represent the linear regression of IP, IC and IP + IC, respectively. The blue straight line represents the regression line, and the pink dotted line and the blue area it contains represent the 95% confidence interval. The dashed orange line and the orange area it contains represent the 95% prediction interval. “r” represents the Pearson correlation coefficient, and “P” denotes the *p*-value.

**Table 1 biomedicines-12-00580-t001:** Patient characteristics for all 197 AIS patients collected.

Characteristics	All 197 Patients
Median age, years (IQR)	72 (63–80)
Gender, male, no. (%)	113 (57.36)
Median Onset-to-CT time (IQR), min	73 (41–180)
Median baseline NIHSS (IQR)	11 (6–17)

IQR: Interquartile Range; NIHSS: National Institutes of Health Stroke Scale.

**Table 2 biomedicines-12-00580-t002:** Comparison of IP and IC segmentation performance with some 2D and 3D methods. The best metric is shown in bold, and the second best is underscored. ↑ denotes that higher values are better and ↓ denotes that lower values are better. All metrics are reported as mean ± std.

Methods	IP			IC		
	DSC (%) ↑	HD95 (mm) ↓	ASSD (mm) ↓	DSC (%) ↑	HD95 (mm) ↓	ASSD (mm) ↓
TransUNet [35]	30.00 ± 19.11	42.22 ± 27.05	10.62 ± 8.19	23.16 ± 16.75	37.27 ± 18.00	9.88 ± 4.06
Swin-UNet [36]	20.99 ± 15.50	39.97 ± 23.35	12.45 ± 8.84	13.97 ± 12.60	53.95 ± 20.15	15.31 ± 7.30
2D nnUNet [9]	31.32 ± 23.73	29.05 ± 24.65	10.34 ± 13.91	25.35 ± 21.11	24.37 ± 15.67	8.64 ± 9.92
UTNet [37]	22.87 ± 15.15	27.13 ± 18.73	7.82 ± 6.65	24.07 ± 21.21	32.36 ± 22.95	9.98 ± 8.71
UNet [38]	28.60 ± 19.98	33.64 ± 25.93	10.16 ± 13.94	13.69 ± 14.79	30.92 ± 19.61	10.14 ± 7.05
V-Net [39]	20.05 ± 18.83	32.93 ± 19.48	10.51 ± 9.11	14.57 ± 15.95	30.19 ± 15.11	10.34 ± 7.26
AttnUNet [25]	34.11 ± 19.93	21.30 ± 14.12	5.91 ± 6.24	27.93 ± 22.26	25.82 ± 19.94	6.92 ± 4.95
UNet++ [40]	33.76 ± 21.32	24.09 ± 19.43	6.13 ± 5.70	27.99 ± 24.49	24.30 ± 18.57	7.11 ± 5.47
3D nnUNet [9]	40.50 ± 21.70	16.50 ± 14.96	4.16 ± 4.08	30.74 ± 23.58	21.17 ± 16.12	6.26 ± 5.65
D-Former [11]	31.46 ± 23.01	23.16 ± 17.24	8.51 ± 9.52	17.89 ± 16.83	25.91 ± 19.09	9.44 ± 13.12
CoTr [13]	36.92 ± 23.93	25.82 ± 21.39	7.23 ± 8.08	17.39 ± 15.51	22.54 ± 11.57	6.77 ± 3.99
TransBTS [41]	26.82 ± 16.47	45.18 ± 22.77	12.06 ± 8.04	19.71 ± 18.33	40.78 ± 21.78	12.61 ± 7.92
UNETR [42]	15.74 ± 10.10	53.37 ± 19.43	15.36 ± 6.23	15.77 ± 19.05	36.66 ± 22.91	13.87 ± 9.07
nnFormer [12]	20.04 ± 15.23	40.71 ± 19.99	12.48 ± 9.86	19.03 ± 22.35	36.00 ± 24.27	13.69 ± 11.03
Swin-UNETR [14]	22.09 ± 14.51	47.61 ± 22.91	13.19 ± 7.72	15.07 ± 15.78	48.29 ± 20.28	16.71 ± 8.62
3D UX-Net [10]	17.78 ± 13.48	41.33 ± 20.91	13.50 ± 10.44	14.82 ± 14.27	32.88 ± 20.19	12.10 ± 7.04
$I^{2}$PC-Net	**42.76 ± 21.84**	**13.81 ± 10.39**	**3.59 ± 2.25**	**33.54 ± 24.13**	**21.02 ± 14.81**	**5.85 ± 4.28**

**Table 3 biomedicines-12-00580-t003:** Comparison of the entire infarct (IP + IC) segmentation performance with some 2D and 3D methods. The best metric is shown in bold, and the second best is underscored. ↑ denotes that higher values are better and ↓ denotes that lower values are better. All metrics are reported as mean ± std.

Methods	IP + IC		
	DSC (%) ↑	HD95 (mm) ↓	ASSD (mm) ↓
TransUNet [35]	44.54 ± 16.24	43.19 ± 25.19	9.63 ± 6.08
Swin-UNet [36]	31.86 ± 18.55	45.22 ± 21.84	12.08 ± 6.77
2D nnUNet [9]	49.44 ± 23.63	24.70 ± 20.33	6.56 ± 6.60
UTNet [37]	40.12 ± 17.44	26.20 ± 18.80	6.41 ± 3.51
UNet [38]	39.94 ± 22.98	33.47 ± 25.34	10.10 ± 13.71
V-Net [39]	31.90 ± 24.15	32.43 ± 19.38	10.23 ± 9.22
AttnUNet [25]	51.48 ± 22.74	20.74 ± 13.39	5.88 ± 6.27
UNet++[40]	51.56 ± 23.48	24.20 ± 19.08	6.22 ± 5.87
3D nnUNet [9]	62.22 ± 11.74	15.26 ± 12.73	3.67 ± 3.64
D-Former [11]	45.18 ± 23.25	22.53 ± 16.57	7.36 ± 8.04
CoTr [13]	50.09 ± 22.14	25.10 ± 20.22	5.89 ± 4.41
TransBTS [41]	39.77 ± 17.94	45.35 ± 21.83	12.09 ± 7.25
UNETR [42]	25.02 ± 13.68	53.73 ± 18.42	15.38 ± 6.01
nnFormer [12]	32.15 ± 19.93	39.93 ± 19.40	11.92 ± 7.90
Swin-UNETR [14]	32.71 ± 17.92	47.63 ± 21.62	13.46 ± 7.61
3D UX-Net [10]	28.45 ± 18.53	39.00 ± 20.29	12.16 ± 7.84
$I^{2}$PC-Net	**65.67 ± 12.30**	**12.54 ± 7.96**	**2.88 ± 1.27**

**Table 4 biomedicines-12-00580-t004:** Classification performance when IP + IC volume is dichotomized using 70mL as cut-off (95%CI is shown in square brackets). * denotes that there is significant difference between two subgroups (*p* < 0.05).

Variable	Subgroup	Kappa	AUC	Accuracy (%)
Gender	female	0.7826 [0.4440–1.0000]	0.9286 [0.8214–1.0000]	0.9000 [0.6990–0.9721]
	male	0.6667 [0.3389–1.0000]	0.8125 [0.6429–1.0000]	0.8500 [0.6396–0.9476]
Age (years)	<70	0.5000 [0.1250–0.8623]	0.7500 [0.5625–0.9286]	0.7500 [0.5050–0.8982]
	≥70	0.9155 ^*^ [0.7143–1.0000]	0.9643 ^*^ [0.8846–1.0000]	0.9583 ^*^ [0.7976–0.9926]
Baseline NIHSS	<9	0.7547 [0.4217–1.0000	0.8333 [0.5000–1.0000]	0.9231 [0.6669–0.9863]
	≥9	0.6897 [0.3969–0.9222]	0.8947 [0.7895–0.9750]	0.8519 [0.6752–0.9408]
Onset-to-CT time (min)	<180	0.6316 [0.3331–0.9041]	0.8462 [0.7083–0.9643]	0.8095 [0.6000–0.9233]
	≥180	0.8939 ^*^ [0.6587–1.0000]	0.9444 ^*^ [0.8125–1.0000]	0.9474 ^*^ [0.7536–0.9906]

## Data Availability

The data presented in this study are available upon request from the corresponding author.

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
