# Peer review of "Segmenting Ischemic Penumbra and Infarct Core Simultaneously on Non-Contrast CT of Patients with Acute Ischemic Stroke Using Novel Convolutional Neural Network"

_biomedicines, 2024, doi:10.3390/biomedicines12030580_

Round 1

Reviewer 1 Report

Comments and Suggestions for Authors

Comments to the Authors:

The manuscript entitled " Segmenting Ischemic Penumbra and Infarct Core Simultaneously on Non-Contrast CT of Patients with Acute Ischemic Stroke Using A Novel Convolutional Neural Network   " is an exciting subject that fits nicely into the journal's scope. It introduces a novel machine learning method in one of the critical medical issues.

This reviewer would like to make some comments to the authors of the proposed paper before it is published.

1.     The reviewer recommends that you summarize the contributions of your proposed paper at the end of the introduction.

Comments on the Quality of English Language

Comments to the Authors:

The manuscript entitled " Segmenting Ischemic Penumbra and Infarct Core Simultaneously on Non-Contrast CT of Patients with Acute Ischemic Stroke Using A Novel Convolutional Neural Network   " is an exciting subject that fits nicely into the journal's scope. It introduces a novel machine learning method in one of the critical medical issues.

This reviewer would like to make some comments to the authors of the proposed paper before it is published.

1.     The reviewer recommends that you summarize the contributions of your proposed paper at the end of the introduction.

2.     In recent years, machine learning and deep learning methods have been extensively used for fully automatic medical image segmentation. Line 39

recommend changing the above sentence to the following:

Machine learning and deep learning methods have been extensively used in recent years for fully automatic medical image segmentation.

3.     There are numerous general 3D medical image segmentation methods available for the segmentation of IP and IC, such as 41 [9–14], etc. Line 40

recommend changing the above sentence to the following:

Numerous general 3D medical image segmentation methods are available for the segmentation of IP and IC, such as 41 [9–14], etc.

4.     Werdiger et al. [20] explored XGBoost followed by 3D neighborhood 50 analysis, for the concurrent segmentation of IP and IC on CTP scans. Line 50

recommend changing the above sentence to the following:

Werdiger et al. [20] explored XGBoost, followed by 3D neighborhood 50 analysis, for the concurrent segmentation of IP and IC on CTP scans.

5.     Sathish et al. [22] employed an adversarially trained CNN to simultaneously segment IP and IC from multi-sequence Magnetic Resonance Imaging (MRI) scans. Line 54

recommend changing the above sentence to the following:

Sathish et al. [22] employed an adversarially trained CNN to segment IP and IC simultaneously from multi-sequence Magnetic Resonance Imaging (MRI) scans.

6.     In summary, the specialized methods mentioned here either do not strictly differentiate between IP and IC or rely on multiple advanced imaging modalities such as CTP, PWI, and DWI for differentiation of IP and IC. However, these advanced imaging techniques are time-consuming and sometimes even unavailable, and fast and cheap Non-Contrast CT (NCCT) is seldom considered in previous studies.

recommend changing the above sentence to the following:

In summary, the specialized methods mentioned here either do not strictly differentiate between IP and IC or rely on multiple advanced imaging modalities such as CTP, PWI, and DWI to differentiate IP and IC. However, these advanced imaging techniques are time-consuming and sometimes even unavailable, and fast and cheap Non-Contrast CT (NCCT) has seldom been considered in previous studies.

Reviewer 2 Report

Comments and Suggestions for Authors

In order to segregate the infarct core and ischemic penumbra of patients with acute ischemic stroke concurrently on non-contrast CT, the authors developed a novel convolutional neural network called I2PC-Net.  Ischemic Penumbra and Infarct Core segmentation is done automatically and simultaneously by I2PC-Net using just Non-Contrast Computed Tomography . About 200 patients were involved in the study. With good localization of affected regions, strong volume correlation, and high dichotomized volume classification performance, findings of this study show that I2PC-Net mayassess ischemic penumbra and infarct core. This could potentially provide valuable information for diagnosis and patient selection in clinical applications and for personalized rehabilitation approach (10.1016/j.clinph.2020.04.158 / 10.26355/eurrev_202309_33580).

The paper is well written and argumented. 

Is this study approved by an institutional board?

Reviewer 3 Report

Comments and Suggestions for Authors

In this manuscript the authors investigated the segmentation of penumbra and infarct core in patients with acute ischemic stroke with non-contrast CT (NCCT). The authors state that they designed a novel convolutional neural network named I2PC-Net which relies solely on Non-Contrast Computed Tomography for automatic and simultaneous segmentation of IP and IC. Their method is said to be widely applicable for automated IP and IC segmentation and desired in computer-aided diagnosis of AIS. Moreover, the authors claimed that their method NCCT is fast and cheap, as compared to others.

The work seems interesting. The manuscript is well written, with the exception of the point mentioned below, that should be improved.

General: is your describe new method accessible to everyone? Please give a statement on this.

Abstract:

Clear

Introduction:

please do not use the word “employed” so often.

Am I wright that you want to show: with your introduced I2PC-Net you can do what others already do, but you can do it faster and cheaper and without advanced imaging techniques?

Material and Methods:

Lines 78: are these numbers right?: 512, 512-638, 28-37

Line 81f: are the authors of the manuscript the mentioned experienced neuroradiologists? Please state clearly.

The Figure 1 legend must be extended: what is green, what is red?

All in all, the technical and IT methods used sound understandable and clear.

Results:

You say: In the figure, we could see that our I2PC-Net could accurately locate affected regions in GTs of IP well and IC and also match the GT labels (DSC=47.44% and 74.57% for IP and IC) better than the 3 compared methods, showing it is poetential to provide affected region information in clinical applications. But where can I find these numbers in Fig 3, where the refer to?

Line 286: You say: As shown in Figure 3, our method could accurately locate the entire ischemic regions and match the GT IP+IC well. But, which of te 6 Figure do you talk about?

Fig 3: the figure is unclear: please clearly indicate, which of the 6 picture is the result from what method – what is the difference between (from left to right) 2rd and 3rd picture?

Fig 4: please describe what is pink and what is blue

Line 299-300: do you have an explanation for the IC relatively weak correlation data?

Discussion:

Line 335f: does it mean that it is better to wait scanning the ill brain at least for 180 min to get better results?

Comments on the Quality of English Language

mostly ok

Round 2

Reviewer 3 Report

Comments and Suggestions for Authors

Dear authors,

thank you for the indented corrections. The manuscript really improved.